# A Review of the Ethnomedicinal Uses, Biological Activities, and Triterpenoids of *Euphorbia* Species

**DOI:** 10.3390/molecules25174019

**Published:** 2020-09-03

**Authors:** Douglas Kemboi, Xolani Peter, Moses Langat, Jacqueline Tembu

**Affiliations:** 1Department of Chemistry, Tshwane University of Technology, Pretoria 0001, South Africa; 2Defense Peace Safety and Security, Center for Science Innovation and Research (CSIR), Pretoria 0001, South Africa; XPeter@csir.co.za; 3Jodrell Laboratory, Natural Capital and Plant Health Department, Royal Botanic Gardens Kew, Richmond TW9 3DS, UK; M.Langat@kew.org

**Keywords:** *Euphorbia*, ethnomedicinal, phytochemistry, triterpenoids, anticancer

## Abstract

The genus *Euphorbia* is one of the largest genera in the spurge family, with diversity in range, distribution, and morphology. The plant species in this genus are widely used in traditional medicine for the treatment of diseases, ranging from respirational infections, body and skin irritations, digestion complaints, inflammatory infections, body pain, microbial illness, snake or scorpion bites, pregnancy, as well as sensory disorders. Their successes have been attributed to the presence of diverse phytochemicals like polycyclic and macrocyclic diterpenes with various pharmacological properties. As a result, *Euphorbia* diterpenes are of interest to chemists and biochemists with regard to drug discovery from natural products due to their diverse therapeutic applications as well as their great structural diversity. Other chemical constituents such as triterpenoids have also been reported to possess various pharmacological properties, thus supporting the traditional uses of the *Euphorbia* species. These triterpenoids can provide potential leads that can be developed into pharmaceutical compounds for a wide range of medicinal applications. However, there are scattered scientific reports about the anticancer activities of these constituents. Harnessing such information could provide a database of bioactive pharmacopeia or targeted scaffolds for drug discovery. Therefore, this review presents an updated and comprehensive summary of the ethnomedicinal uses, phytochemistry, and the anticancer activities of the triterpenoids of *Euphorbia* species. Most of the reported triterpenoids in this review belong to tirucallane, cycloartanes, lupane, oleanane, ursane, and taraxane subclass. Their anticancer activities varied distinctly with the majority of them exhibiting significant cytotoxic and anticancer activities in vitro. It is, therefore, envisaged that the report on *Euphorbia* triterpenoids with interesting anticancer activities will form a database of potential leads or scaffolds that could be advanced into the clinical trials with regard to drug discovery.

## 1. Introduction

The family *Euphorbiaceae*, also known as the spurge family, is one of the largest families of flowering plants conspicuously throughout the tropics and composed of over 300 genera and 8000 species [1,2]. The family is very diverse in range, distribution, and morphology and is composed of diverse species that grow on main land as small/large shrubs, small/large weeds, and large/woody trees, or as climbing lianas [2]. *Euphorbia* is among the largest genera of flowering plants reported in this family *(Euphorbiaceae*) and contains several other subgenera and sections with over 2000 species [3]. The species in this genus, and to a larger extent of the spurge family, are characterized by the production of a milky irritant latex [1]. The genus consists of species of great economic importance, which makes it a complex genus with great research potential.

Their successes in traditional medicine usage have been attributed to a wide variety of chemical compositions of its members, such as essential oils, oxygenated sesquiterpenes, sesquiterpenes hydrocarbons [4], as well as the macrocyclic diterpenoids [3]. As a result, these chemical constituents are now providing lead compounds for drug discovery because of their therapeutic applications, such as cytotoxic, antitumor, antiviral properties, multi-drug-resistance-reversing, and anti-inflammatory activities as well as their great structural diversity, resulting from various polycyclic and macrocyclic skeletons, including but not limited to jatrophane, ingenane, daphnane, tigliane, lathyrane [3].

Conversely, the structural diversity of *Euphorbia* triterpenoids has also been reported. The pharmacological activities of these triterpenoids have been associated with the important functional groups on the ring. Indeed, the various pharmacological activities reported for some pentacyclic triterpenoids like anticancer, have been attributed to the presence of the hydroxyl group. This was evident by racemosol, a polyhydroxy triterpenoid that showed significant anticancer activities against human cancer cells [5]. Other studies have further shown that the activities of hydroxylated pentacyclic triterpenoids are significantly affected by the number of hydroxyl groups contained in ring A. This was observed on chemically modified 18β-Glycyrrhetinic acid, an oleanane-type triterpene [6]. Furthermore, a highly hydroxylated and bioactive pentacyclic triterpenoid named 1β, 2α, 3β, 16β, 19β,)-1, 2, 3, 16, 19,pentahydroxyolean-12-en-28-oic acid from *E. sieboldiana* exhibited modest anticancer activities against human liver carcinoma (HepG2) cells, human cervical cancer cells (HT-3), and normal human carcinoma (HeLa) cells [7]. The observed anticancer activities of this triterpenoid were attributed to the presence of tetra hydroxyl groups in ring A [7]. Plants derived terpenes have further been associated with therapeutic effects like anticancer and anti-inflammatory [8], as well as the ability to overcome drug resistance, and cancer migration [9]. This is due to their ability to block the NF-κB activation, induce apoptosis, and inhibit proliferation as exhibited by a reduced lantadene A triterpenoid [10] and rosamultic acid against human carcinoma [11]. Undeniably, plants are known to biosynthesize complex and diverse triterpenes to protect them against predators like herbivores and against pathogens and pests [12].

Therefore, triterpenoids can provide potential leads that can be developed into pharmaceutical compounds for a wide range of medicinal applications. However, there is scattered scientific reports about the anticancer activities of these constituents. Furthermore, there is lack of consolidation of the latest information and discoveries pertaining the anticancer properties of the isolated triterpenoids. Harnessing such information can provide a database of bioactive pharmacopeia or targeted scaffolds that can be advanced into more accessible and economical drugs with fewer side effects on normal cells.

In addition, several reviews on *Euphorbia* species have mainly reported their pharmacological activities [13], ethnomedicinal usage [14], and the structural diversity of isolated diterpenoids [3,15] as well as essential oils [4]. Hence, in order to promote more targeted future research of this species, consolidation of scientific reports/evidence of their ethnomedicinal uses, the chemical constituents, and the identification of possible knowledge gaps need to be addressed. In view of this, the current review reports the ethnomedicinal uses and triterpenoids isolated from *Euphorbia* species, highlighting their anticancer activities and areas for future studies. Therefore, it is envisaged that *Euphorbia* triterpenes will offer interesting scaffolds that can be used for drug discovery. It is also our hope that this review will provide a resourceful reference for the necessary information that will inform future studies on the therapeutic potential of *Euphorbia* species and their constituents.

## 2. Literature Survey Databases

In order to investigate the ethnomedicinal uses, biological activities, and isolated triterpenoids of *Euphorbia* species, an extensive online literature search was conducted using terms like ‘*Euphorbia* medicinal uses’ and ‘*Euphorbia* triterpenoids’ on the online databases such as Scifinder, Springer Link, Scopus, Wiley online, PubMed, science Direct, and Google Scholar. The search resulted in over 900 reports mainly of the English language, which we accessed at our institution. The reports contained information of early documents and reviews on *Euphorbia* uses dating back to ancient times as well as current reports. The retrieved information was conceptualized, carefully read, critically analyzed searched for explanations of medicinal uses of *Euphorbia* species, their biological activities, isolated triterpenoids, and their pharmacological properties. Correct plant names were further confirmed from the plant list. Hence, the results presented in the current review are interpreted, analyzed, and documented only in the context of the literature information obtained.

## 3. Ethnomedicinal Uses of *Euphorbia* Species

The medicinal uses of *Euphorbia* species are very diverse. The reported literature indicates that most species are used in the treatment of diseases, ranging from respirational infections, body and skin irritations, digestion complaints, blood syndromes, inflammatory infections, body pain, genitourinary syndromes, microbial illness, snake or scorpion bites, cellular tissue ailments, nutritional diseases, injuries, endocrine, pregnancy/birth/puerperium disorders, and sensory disorders [13,14,15,16,17]. Reports also show that their medicinal application is recorded all over the world and utilizes different plant parts such as whole plant, latex, aerial part, leaves, stems, and roots. However, *E. hirta* is the most commonly used species for treatment of a diverse range of diseases [14,16]. Others include *E. terracina*, *E. biumbellata* [17], as well as *E. lathyris*.

In digestive systems, *E. hirta* preparations are used for the treatment of female disorders, worm infestation in children, pimples, gonorrhea, tumors, constipation, dysentery, jaundice, indigestion, biliousness, stercorrhagia, bloating, choleric effects, and the laxative [18]. For instance, a decoction of leaves or stems of *E. hirta* and *E. lathyris* is often administered as anti-diarrheal in Burundi, Philippines, China, and Nigeria [18,19]. *Euphorbia* species have also been recognized and reported for their anticancer and anti-hepatitis B components [19].

As skin remedies, the preparations from *Euphorbia* species are used for alleviating skin itching, warts, eczema, hair loss, acne, dermatitis, boils, sunburn, calluses, rashes, irritation, pustules, and for antiseptic, disinfectant, and emollient properties [14]. The most commonly used species for this category of infections is *E. maculata*, *E. hirta* [19], *E. peplus*, *E. sessiliflora*, *E. apios*, and *E. macroclada* among others.

Many scientific publications have also described the use of *Euphorbia* species for treatment of microbial infections, including malaria, cancer [19], ringworms, anthelmintic, tuberculosis, and sexually transmitted diseases such as syphilis and gonorrhea [14,19,20]. In addition, the latex of *Euphorbia* is a strong purgative and vesicant [20] and has been used for the treatment of lice infestations, scabies together with the treatment of mange, a skin disease affecting domestic animals. The treatment of parasitic infections like anthelmintic and measles have also been reported. *E. hirta*, *E. humifusa*, *E. nivulia*, and *E. seguieriana* are the most cited in this category.

Also, other literature records have described the use of *Euphorbia* species in the treatment of respiratory illness such as asthma, coughs, bronchial complaints, and pneumonia [14,18]. Indeed, the extracts from *Euphorbia* species have further shown pharmacological activities like anti-arthritis, anti-inflammatory, antioxidant, antitumor, antisplasmodic, and anti-proliferative properties in vitro and these extracts have been isolated and patented as modern drugs [21]. These usages have been reported in most countries with *E. hirta* as the most cited species together with *E. pululifera*, *E. neriifolia*, and *E. rigida* [14].

Besides, other records have documented the use of *Euphorbia* species in the treatment of inflammation associated with the digestive system, such as dysentery as well as respiratory system illness, genitourinary inflammation, and sensory systems disorders [21]. Decoctions of leaves of related families like *Lamiaceae* and *Asteraceae* are also used to treat respiratory system illness like common colds [22]. Again *E. hirta* was reported as the most used species in managing these infections [19,23]. The use of *E. hirta* is prehistorical as it was used for the treatment of respiratory diseases like bronchitis by Nicander, a Greek physician in the 2nd century before Christ (BC) and has also been described for the same usage in Australia, Hawaii [23], and South-western USA.

*Euphorbia* species have further been reported to treat other abnormalities, including edemas, swellings, dropsy, fistula, lesions, as well as injuries such as wounds, abscesses, blisters, and burns. Several species have been cited in this category and include *E. tirucali*, *E. ingens*, *E. apios*, *E. neriifolia*, *E. macraclada*, *E. nicaeensis*, *E. coniosperma*, and *E. sikkimensis*, among others. They have further been cited for their use in the management of genitourinary system disorders, including hematuria, diuretics, emmenagogues, aphrodisiacs, leucorrhoea, chyluria, as well as male and female infertility [14]. Other usages are for treatments of aphrodisiacs oliguria, impotence, dysuria, blennorhagia, bladder stones, and chyluria [19,23]. *E. sessiliflora*, *E. lagascae, E. prostrata*, *E. maculata* and *E. pekinensis* were predominant in these categories.

Furthermore, *Euphorbia* species are used for relieving pain such as chest pain, skeletal pain, toothache, stomachache, migraines, and headache. The uses of *Euphorbia* preparations and decoctions for the treatment of dry skin conditions like dermatitis, calluses, in addition to eye disorders have further been documented [14,21]. *Euphorbia* species have also been mentioned for their usage by mothers as lactation stimulants and as labor inducers and for cleansing or expelling the placenta. The most quoted species in this category was *E. thymifolia* alongside other unnamed species [14]. Other species have been used for the treatment of eye disorders, including those associated with cataracts, a condition that causes blindness and deafness. In Southern California, decoctions/preparations of *E. maculata* is used for the treatment of warts and blindness by removal of corneal opacities [24]. The use of *Euphorbia* species to treat poisoning associated with snake bites and bees or scorpions’ stings have also been reported [14,23].

Other medicinal uses of *Euphorbia* species have been described in the categories of social uses, environmental uses, neoplasms, ornamentals uses, fuelwood, timber products, and wood crafts, hedge/fence [14,21]. Other miscellaneous uses include the treatment of rheumatism conditions, as nutrients in food, as blood purifiers for paralysis and hallucination disorders [14]. The most utilized species in these categories are *E. ingens*, *E. antiquorum*, *E. fischeriana*, *E. esula*, *E. lathyris*, *E.serpens*, *E. nivulia*, and *E. peplus* [14,23].

However, despite most of the *Euphorbia* species considered as toxic, some species have been used as food. Documented reports describe the use of leaves, stems, roots, and in some cases, the whole plant being eaten as food for humans and animals. For instance, in Spain, the fresh latex of native *E. serrata* is used to hasten the process of coagulating milk, while in Turkey, the latex of *E. stricta* and *E. amygdaloides* is used to clean water for domestic uses [14]. Conversely, milk contamination has been reported from cattle that are fed on some *Euphorbia* species. It is also believed that the toxic properties of honey could be enhanced by the addition of some *Euphorbia* species [23]. The latex and the roots are the most common used parts. Others include seeds, flowers, stem wood, and stem barks, leaves and whole plants, as indicated in Table 1 alongside their curative properties.

The above accounts show the diverse medicinal uses of *Euphorbia* species. This is attributed to the fact that *Euphorbia* species are widely distributed in nature with varying climatic and soil disparities, hence they tend to manufacture a wide range of unusual secondary metabolites to aid their response to these disparities in the habitats [21]. The various types of chemical constituents of these species makes most of the members therapeutically relevant and also poisonous [25,26,27].

Hence, *Euphorbia* species are exceptionally valuable sources of biologically active natural products that could be utilized in the treatment of various diseases. However, even though most of the *Euphorbia* species are cited in folk medicine, their dosage, mechanisms, mode of actions, side effects, and efficacy are not clear and well elaborated. Hence, there is a need for more comprehensive research to establish their safety and efficacy.

## 4. Isolated Triterpenoids

The phytochemistry of *Euphorbia* genus is complex with a variety of chemical constituents of different classes. Among them, the structural diversity of cyclic triterpenes is incredible. At least over 130 different types of triterpenoids have been isolated to date from different *Euphorbia* species. Reported evidence shows that tetracyclic triterpenoids are the major triterpenoids in *Euphorbia* species and occur in three main classes; tirucallane (**1**–**19**), euphane (**20**–**39**), lanostane (**85**–**94**), and cycloartanes (**40**–**84**). Cycloartanes (9,19-cyclolanostanes) are some of the main tetracyclic triterpenes that contain the side chain and a characteristic cyclopropane ring. They are the major key intermediates in the phytosterols biosynthesis and are used as specific chemotaxonomic markers in *Euphorbia* genus.

Besides, pentacyclic triterpenoids such as lupane (**95**–**101**), oleanane (**102**–**118**), taraxarane (**119**–**124**), friedoursane, friedelane (**125**–**126**), and ursane triterpenoids have also been reported in the genus. However, only a few friedoursane triterpenoids have been reported from *Euphorbia* species to date. Table 2 highlights the structural diversity of *Euphorbia* triterpenoids alongside Figure 1, Figure 2, Figure 3, Figure 4 and Figure 5.

In addition, other unusual pentacyclic triterpenoids, such as eupulcherol A, with an uncommon carbon skeleton, have been isolated from *E. pulcherrima* extracts of the whole plant. It was observed that the AB ring intersection without the methyl groups is very uncommon, though it is common in some hopanes from oils and sediments [40]. Also, Ricardo et al. [53] described the isolation of a tetracyclic triterpenoid with an unusual spiro scaffold for the first time, which was named spiropedroxodiol. While Chao et al. [54] reported for the first time, the isolation of two nor-triterpenoids from the root extracts of *E. ebracteolata*, which were named ebracpenes A and B. Ebracpenes A was found to possess a rare C-seco ring probably occasioned through cleavage reactions, while ebracpenes B possessed an additional uncommon aromatic ring D.

In a similar way, three oleanane type triterpenoids; euphorimaoid A and euphorimaoid B, were reported from the aerial parts of *E. pulcherrima* [55]. Other unique nor-triterpenoid isolated from *E. soongarica* and, which shared a tetracyclic ring with a euphane triterpenoid is (+)-(*24S*)-eupha-8,25-diene-3β,24-diol-7-one (**32**). However, they only differ in the C-17 side-chain moiety. Sooneuphanone D and soonoleanone were also isolated from the root extracts of this species [56]. Likewise, the investigation of *E. alatavica* by Rushangul et al. [57] resulted in the isolation of nine compounds, including alatavolide and alatavoic acid reported for the first time.

Among the triterpenoids isolated from *Euphorbia* species, over 10 different skeletal structures have been identified. Tirucallane and euphane triterpenoids are most common and are specific to the genus, while cycloartane, lanostane, and other pentacyclic triterpenoids are not specific to the genus as they have been isolated in other plant families. In higher plants, triterpenes are believed to be biosynthesized via the mevalonate pathway in the presence of squalene synthase enzyme [12].

Their cyclic skeletal structures are formed from an enzymatic reaction using (3*S*)-2,3-oxidosqualene as the precursor. The cyclization of squalene is through squalene-2,3-oxide an intermediate produced in a flavoprotein catalyzed reaction in the presence of O_2_ and nicotinamide adenine dinucleotide phosphate (NADPH) cofactors [12]. This result into terminal bond formation in the squalene. The squalene subsequently cyclizes into different conformations, such as chair-chair-chair-boat to produce various skeletal structures.

If the squalene-2,3-oxide is folded into a chair-chair-chair-boat conformation on the enzyme surface, an intermediate dammarenyl cation is produced. A series of cyclization of the cation followed by a chain of Wagner-Meerwein rearrangements of 1,2-hydride shift, deprotonation and 1,2-methyl migrations leads to formation of different triterpene scaffolds of various skeletal structures such as euphane, tirucallane, lanostane, as well as other tetracyclic triterpenoids.

Consequently, the 1,2-alkyl shift in the dammarenyl cation leads to the formation of bacharenyl cation, which undergoes further cyclization, ring expansion, hydride shifts, rearrangement, and methyl shifts to form various skeletal intermediates of pentacyclic triterpenoids like lupane, oleanane, ursanes, taraxanes, and friedelane [12]. Oxidation, aliphatic/aromatic esterification, acetylation, and hydrogenation of this scaffolds produces other diverse titerpenoid skeletal structures.

As a common feature, the biosynthesis of these triterpenoids follows a typical path of C-C ring formation followed by rearrangement or methyl shift reactions. These reactions result in the generation of diverse scaffolds with many stereocenters. Friedelin triterpenoids are examples of the highly rearranged triterpenoids in the genus. Others form alcohols via the incorporation of a hydroxyl group on the C-3 (3β-OH) and the formation of a double bond. Oleanane, ursane, and taraxanes are common examples of triterpenoids bearing a hydroxyl group at the C-3 and a double bond in the ring. In contrast, other plant species synthesize different types of unusual triterpenes scaffolds through uncommon reactions like the introduction of a ketone, formation of an oxide bridge, and ring cleavage. Ebracpenes A and B triterpenoids isolated from *E. ebracteolata* are good examples of nor-triterpenoids, which have undergone ring cleavage, resulting into the formation of a rare C-seco ring. Others are oleanane type triterpenoids; euphorimaoid A and euphorimaoid B from the aerial parts of *E. pulcherrima* [55]. These triterpenoids possess an epoxy, acetyl, and a hydroxyl side chain on the C-3, which are thought to enhance their biological activities.

As for the traditional uses of this species, no particular class of triterpenoid has been identified to a certain morphological group or linked precisely to a known medical condition. However, tirucallane, euphane, and cycloartanes triterpenoids have been identified and isolated in most of the species utilized for medicinal purposes, such as *E. hirta*, *E. thymifolia*, and *E. mili*; hence, it could be considered as the marker compound of the genus. In contrast, the highly hydroxylated pentacyclic triterpenoids, such as oleanane which are known to confer various pharmacological properties have been isolated and identified in only a few of *Euphorbia* species.

## 5. Biological Activities

Considering the traditional uses of *Euphorbia* species, particularly for the treatment of inflammation, skin cancer, and wounds, there seems to be strong indications that the isolated compounds would exhibit associated biological properties as well. Indeed, plant phytochemicals have been reported to be useful for both preventive and therapeutic purposes, and are able to change the differentiation status of some cell types [81].

### 5.1. Anticancer Activities

Emerging evidence has further demonstrated that *Euphorbia* species exhibit various pharmacological activities such as anticancer properties in vitro because of the rich production of bioactive constituents [3]. This supports their utilization in traditional folk medicine as anticancer agents. These constituents are thought to be cytotoxic via different mechanisms of action such as cell proliferation and differentiation, apoptosis and inhibition of metastasis, excessive production of reactive oxygen, and effects on angiogenesis [81].

#### 5.1.1. Effects on Cell Proliferation and Differentiation

As a common feature, cancer undergoes uninhibited simple differentiation and proliferation of cells resulting in complex and abnormal cell division, progression, and multiplication [82]. Hence, an effect on this change is a better strategy in finding cancer treatment. Studies on the ability of kansenone (**27**) triterpenoid, isolated from *E. kansui* to express suppression of cell proliferation were performed by Fangfang et al. [83]. From the results, kansenone (**27**) not only exhibited significant anticancer activities against the cancer cells but also showed cell arrest at the G0/G1 phase. It was further demonstrated to damage the mitochondria as well as up-regulating the apoptotic and anti-apoptotic proteins [83]. From the observations, they concluded that kansenone (**27**) causes cell death (apoptosis) by inducing damage to mitochondria and cell receptor death.

Similarly, the thiazolyl blue tetrazolium bromide (MTT) assay of triterpenoids from *E. pedroi* against human colon adenocarcinoma cells (Colo320), the sensitive cell line (Colo205), and the human embryonic fibroblast cell lines (MRC-5) was investigated and reported. While no appreciable activity (IC50 > 10 μM) was observed for all of the tested triterpenoids (**61**, **62**, **103**–**105**), oleanolic acid (**106**) exhibited better activity against resistant mouse lymphoma cells. Also, cycloart-25-ene-3β-24-diol (**63**) showed promising cytotoxic activities against MRC-5, as compared to standard cells [53].

#### 5.1.2. Cytotoxic Effects

The cytotoxicity effects of a given targeted therapy against cancer cells can be achieved through several pathways. Cycloartanes tetracyclic triterpenoids, including; 25-hydroperoxycycloart-23-en-3β-ol (**67**), cycloartenol (**74**), 24-hydroperoxycycloart-25-en-3β-ol (**76**) and taraxerone (**120**) isolated from *E. hirta* were evaluated for their cytotoxicity against colon carcinoma (HCT 116), a human cancer cell line. The triterpenoid 25-hydroperoxycycloart-23-en-3β-ol (**67**) and 24-hydroperoxycycloart-25-en-3β-ol (**76**) showed good activity with an IC50 value of 4.8 μg mL^−1^, while taraxerone (**120**) and cycloartenol (**74**) were inactive against HCT 116. However, **67** and **76** were tested for cytotoxicity against non-small cell lung adenocarcinoma (A549) with an IC50 value of 4.5 μg mL^−1^ [72]. In addition, taraxast-12-ene-3β, 20, 21(α)-triol, cycloartane-3β, 25-diol (**119**), and cycloartane-3β, 24, 25-triol isolated from *E. denticulate* exhibited cytotoxic effects, with IC50 values of 12.2 ± 2.9, 27.5 ± 4.9, and 18.3 ± 1.4 μM, against prostate cancer cells, respectively [79]. Furthermore, eupha-8, 25-diene-3, 24-diol-7,11-dione (**5**), eupha-8,24-diene-3β,11β-diol-7-one (**6**), and eupha-8-ene-3β,11β-diol-7,24-dione (**7**), isolated from *E. resinifera*, exhibited various degrees of cytotoxic effects against human breast adenocarcinoma (MCF-7) and cellosaurus (C6) cancer cell lines. However, (−)-(24*R*)-tirucalla-8,25-diene-3*β*,24-diol (**8**) displayed the highest potency against MCF-7 and C6, with IC50 values of 56.2 μM and 49.6 μM, respectively [60]. Later on, the cytotoxic activities of triterpenoids (**49**, **50**, **53**) from *E. schimperi* were evaluated using the sulforhodamine B (SRB) assay against MCF-7, HepG2, and HCT-116 cancer cell lines. Cycloart-25-en-3-one (**53**) and 26,27-dinor-3β-hydroxy cycloartan-25-al (**50**) showed the highest cytotoxic activities with IC50 values of 0.18 μM and 0.60 μM, respectively, when compared to an IC50 value of 0.20 μM for doxorubicin [67].

Furthermore, cytotoxic studies of cycloschimperols A (**49**); [cycloart-20,24-dien-3β-ol], B (**50**); [26–27-dinor-3β-hydroxycycloartan-25-al], 24-methylenecycloartane and cycloart-25-en-3-one (**53**) against similar cell lines (MCF-7, HepG2, HCT-116) was conducted by Mohamed et al. [67]. The results revealed that cycloschimperols B (**50**) recorded the highest activity with an IC50 value of 0.60 μM, while cycloart-25-en-3-one (**53**) had the lowest activity with an IC50 value of 0.20 μM in comparison to doxorubicin as the standard [67].

Additionally, the euphorol titerpenoids (**3**, **5**–**7**, **12**, and **14**) from the methanol extracts of *E. resinifera* showed modest activities when evaluated against human myeloid leukemia (U937), MCF-7 and C6 cancer cell lines. Comparatively, the tirucallane triterpenoids (**8**–**11**, **13**, **15**, and **18**) and euphane triterpenoids (**31**, **32**) from *E. micractina* showed paltry activities against human ovarian cancer cells (A2780) in comparison to the standard cells. Likewise, among the cycloartanes (**70**–**73**) isolated from *E. pterococca* whole plant extracts, cycloartenyl-2′E, 4′*E*-decadienoate (**70**) showed selective inhibition of α/β-hydrolase 12 with an IC50 of 11.6 ± 1.9 μM [68].

#### 5.1.3. Apoptosis, Cell Cycle Arrest, and Autophagy

Apoptosis is an orderly event within a biological system, which results in cell death, commonly referred to as ‘cellular suicide.’ It is an important process that looks for damaged cancerous cells in the body [84]. On the other hand, the cell cycle is a series of ordered events that ultimately lead to cell growth and division to other new daughter cells [85]. Conversely, autophagy, unlike apoptosis, is a safer way of clearing out damaged cells in the body so as to regenerate healthier ones [86]. Biological studies on (19αH)-lupane and (9βH)-lanostane isolated from *E. helioscopia* [77] against the HeLa cell lines revealed that they caused cell death by stimulation of cell natural death/apoptosis with significant EC_50_ values of 1.59 ± 0.25 μM for (19αH)-lupane and 26.48 ± 0.78 μM for (9βH)-lanostane [77].

Furthermore, the anticancer screening of new tetra-hydroxylated triterpenoid (1**27**–**130**) isolated from *E. sieboldiana* was studied. The results revealed that a pentacyclic triterpenoid (**127**) inhibited growth of HeLa cells with reduced toxicity. Cell growth and cell proliferation inhibition observed for this compound was attributed to the G1 phase arrest. It was also found to cause stimulation of apoptosis in cervical carcinoma. Further studies showed that the compound deactivates the tumor necrosis factor-alpha induced IKK phosphorylation NF-kappa B (TNF-α–TAK1–IKK-NF-κB) axis, and further inhibited and regulated the NF-κB target genes involved in cell death and cell growth. The effect of this compound on NF-κB of the test cells was; however, found to be through the production of reactive oxygen species (ROS). The destruction of migratory HeLa cells was further deduced from the observations made. From the findings, they concluded that since the compound shows potential anticancer activities, it could be a good anticancer agent [7].

#### 5.1.4. Effects on Drug-Resistant Cancer Cells

Drug resistance occurs mostly in microbial pathogens, where in some way, they (microorganism) reduce the effectiveness of a given drug. Certain proteins commonly recognized as cluster of differentiation 243 (CD243) are known to cause multidrug resistance (MDR). This has been the reason behind the failure of many cancer therapies. However, the glycoprotein (P-gp or Pgp), which is normally programmed with multidrug gene 1 (MDR1) pumps out of the cell any foreign materials/substances [87]. Therefore, targeting compounds that can inhibit these proteins can help in addressing drug resistance and also reduce the migration of cancer cells.

The cytotoxicity and multidrug resistance (MDR) reversal activity of kansuinone (**29**) and sooneuphanone showed moderate cytotoxicity against the KB and KBv200 cell lines [56]. While the compounds alatavoic acid (**121**), 3,2-seco-taraxer-14-en-3,2-lactone (**117**), and 3-hydroxy-4,4,8,14-tetramethyl-18-norpregnan-20-one isolated from *E. alatavica* exhibited potent cytotoxicity. However, 3-hydroxy-4,4,8,14-tetramethyl-18-norpregnan-20-one exhibited highest potent cytotoxicity against A549, HeLa and MCF-7 cells, with IC50 values of 6.5 ± 3.1 μM, 1.9± 0.9 μM, 8.6 ± 3.5 μM, respectively, better than the positive control doxorubicin (DOX) (IC50 values: 4.2 ± 0.3 μM, 7.2 ± 2.6 μM, 15.7 ± 4.2 μM), while the compound alatavoic acid (**121**) showed better cytotoxicity than other isolated compounds, with IC50 values of 16.4 ± 3.2 μM [57].

Similarly, among the triterpenoids (**62**, **63**, **66**, **68**, **69**) isolated from *E. macrostegia*, cycloart-23(*E*)-ene-3β, 25-diol (**62**) was the most active compound on the MDA-MB468 cell line (LD50 = 2.05 μg mL^−1^) and cycloart-23(*Z*)-ene-3β, 25-diol (**66**) was the most active compound on MCF-7 cell line (LD50 = 5.4 μg mL^−1^) [71]. Furthermore, IC50 values for navelbine in combination with triterpenoid sooneuphanone from *E. soongarica* showed an appreciable reduction in a dose-reliant effect. This was a good indicator that the compound possesses promising MDR reversal properties in comparison to verapamil (standard drug) [56].

Other studies have shown that lupenone (**100**) had an anticancer activity against breast cancer cells MCF-7 with an IC50 of 8.07 μg/mL using the neutral red assay [88]. Likewise, lupenone (**100**) inhibited the MCF-7 cells activity, which was consistent with the previous studies. In addition, it showed significant inhibition of tyrosinase with IC50 values of 71.4 μM [89]. Nevertheless, further studies revealed that the compound excites the melanogenesis in B16 2F2 and B16 melanoma cells through the activation of tyrosinase enzyme expression. This expression occurs through the nitrogen-activated protein kinase phosphorylated extracellular signal-regulated kinases phosphorylation [89]. In addition, *E. ebracteolata* triterpenoid, ebracpenes B displayed modest activity with an IC50 value of 0.89 μM against lipase. On further analysis, it was found to exhibit competitive inhibition for binding sites of lipase [54].

Thethiazolyl blue tetrazolium bromide (MTT) assay of cycloart-23(*E*)-ene-3β- 25-diol (**62**) exhibited significant activity on MDA-MB468 cell line (LD50 = 2.05 μg mL^−1^), while cycloart-23(*Z*)-ene-3β,25-diol (**63**) was the most active compound on the MCF-7 cell line (LD50 = 5.4 μg mL^−1^). [71]. The cytotoxic activities of compounds from *E. resinifera* against MCF-7, U937, and C6 cancer cell lines, using paclitaxel as a positive control also revealed that 11*β*-hydroperoxyeupha-8,24-diene-3*β*-ol (**37**) displayed the highest potency against MCF-7, U937, and C6 with IC50 values of 37.36, 47.17, and 46.89 μM, respectively. These observations suggest that introducing a hydroperoxy group in the structure of 11*β*-hydroperoxyeupha-8, 24-diene-3*β*-ol (**37**) may inhibit its cytotoxic activity [59].

In 2011, Sidambaram et al. [90] studied the anti-proliferative activities of the methanol extract of leaves of *E. hirta* against HepG2 cells from the human epithelioma of the larynx. The results revealed that the dose of 625 μg/mL of methanol extract showed anti-proliferative activity. The free radical, superoxide anion, and hydroxyl radical scavenging activity using 1,1-diphenyl-2-pycrylhydrazyl (DPPH) and cytotoxic activity against the human hepatoma cell line (SMMC-7721), human cervix epitheloid carcinoma cell line HeLa, and the human gastric cancer cell line (SGC-7901) of acetone root extracts of *E. hylonoma* was also investigated. The results showed high antioxidant activities for acetone extract and a fraction of ethyl acetate.

### 5.2. Other Biological Activities

Evaluation of isolated triterpenoids against intestinal lipases was also investigated. For example, the nor-triterpenoids named ebracpenes A and B from root extracts of *E. ebracteolata* showed modest lipase inhibition properties with the highest IC50 of 0.89 μM recorded for ebracpenes B [54].

In addition, a triterpenoid named eupulcherol A isolated from *E. pulcherima* was assayed for its anti-Alzheimer activity, a neurodegenerative disorder. The results revealed that the compound had promising anti-Alzheimer’s disease (AD) activities and that it can inhibit the paralysis of transgenic Caenorhabditis elegans. This explains why the species were traditionally used for the treatment of hypermenorrhea, bruises, and traumatic hemorrhage [40].

Anti-influenza activities of *Euphorbia* constituents have also been reported. For instance, a novel anti-influenza molecule 1,3,4,6-tetra-*O*-galloyl-β-Dglucopyranoside, was identified from the extracts of *E. humifusa*. The identified compound was then evaluated by measurement of its inhibition ability of the influenza viruses by utilizing a high-throughput image assay method. The results showed that the molecule possessed anti-influenza activities against seasonal influenza A strains as well as seasonal influenza B strain. Further studies revealed that it inhibits the nuclear export of influenza nucleoproteins (NP) associated with early stages of the infection [35].

The biological evaluation of isolated compounds from *E. stracheyi* for antiangiogenic properties showed no activity for all the tested compounds. Nonetheless, 3-*O*-(2′*E*,4′*Z*-decadienoyl)-20-*O*-acetylingenol (1 μg/mL) killed all the embryos it was tested against. While for those that survived, pericardial edema was witnessed when they were treated with a reduced concentration of the test compound [47]. In addition, compounds from *E. helioscopia* have been reported to exhibit good anti-inflammatory inhibition activities in murine microglial cellosaurus (BV-2) cells which are associated with inflammation disorders. This study relates to usage of this species in traditional medicine for treatment of anti-inflammatory infections. Hence, the species could be used for the discovery of anti-inflammatory drugs [50].

Accordingly, the seven carolignans from the *E. sikkimensis* were investigated for their anti-HIV activities in vitro using zidovudine (AZT) as the positive control. Carolignans A and B showed moderate anti-HIV activity with EC50 values of 6.3 and 5.3 μM. However, one pair of enantiomers, (−)-(7′*R*,8′*S*)-erythro-7′-methylcarolignan E and (+)-(7′*S*,8′*R*)-erythro-7′-methylcarolignan E, showed more potent anti-HIV activity [47].

## 6. Conclusions and Recommendations

In this review, we report the ethnomedicinal, phytochemistry, and anticancer studies of *Euphorbia* species, with an emphasis on triterpenoids and their anticancer properties. Traditionally, *Euphorbia* species have been used for the management of several diseases and ailments ranging from infectious to chronic diseases such as cancer. Their successes have been attributed to the chemical diversity of their isoprenoid constituents such as triterpenoids. As a result, several studies have reported the isolation of different classes of triterpenoids with the majority of them being taraxanes, euphane, tiricallane, cycloartanes, oleanane, and some with rare skeletal structures such as eupulcherol A. This enriches the structural diversity of the *Euphorbia* species. These triterpenoids have also been demonstrated to be cytotoxic via different mechanisms of action, such as cell proliferation and differentiation, apoptosis, and inhibition of excessive metastasis production of reactive oxygen and effects on angiogenesis.

The biological studies reviewed in this report employed mostly in vitro studies, while some conducted in vivo studies using rodent models. Hence, the studies cannot be extrapolated to humans. Most findings from these studies reported promising anticancer activities of the isolated triterpenoids with high IC50 values. The pharmacological activities of these triterpenoids have further been associated with important functional groups on the ring. Indeed, the biological activities of highly hydroxylated pentacyclic triterpenes have been associated with the number of hydroxyl groups in the ring. Hence, hydroxylated triterpenoids could be promising anticancer agents for cancer treatment.

However, some studies reported no significant inhibitory activity against cancer cell lines. Furthermore, very little has been achieved in advancing these studies to clinical trials. For the many isolated triterpenoids, it is surprising that little has been done in the evaluation of their toxic effects, which is of prime importance for any drug. In addition, from the reported information, it is evident that some of the isolated triterpenoids for the first time have not been evaluated yet could be potential lead compounds for drug discovery. Therefore, for an in depth analysis of the medicinal relevance of *Euphorbia* species, detailed and extensive in vitro and in vivo studies of isolated compounds, their mode of action, and their toxicities against a wide range of pathogens need to be conducted. Furthermore, the utilization of advanced throughput screening of *Euphorbia* plants extracts will possibly explain the observed synergisms, as well as other mechanisms of actions of this natural products from which better new drugs will be discovered and inspired. This will also improve the available knowledge about the phytochemistry, chemical constituents, toxicity, and efficacy of the bioactive compounds, which can give scientific credence to the ethnomedicinal use of *Euphorbia* species. In addition, the structural diversity exhibited by some rare compounds requires further studies about their potential bioactivities and their chemotaxonomic roles.

## Figures and Tables

**Figure 1 molecules-25-04019-f001:**
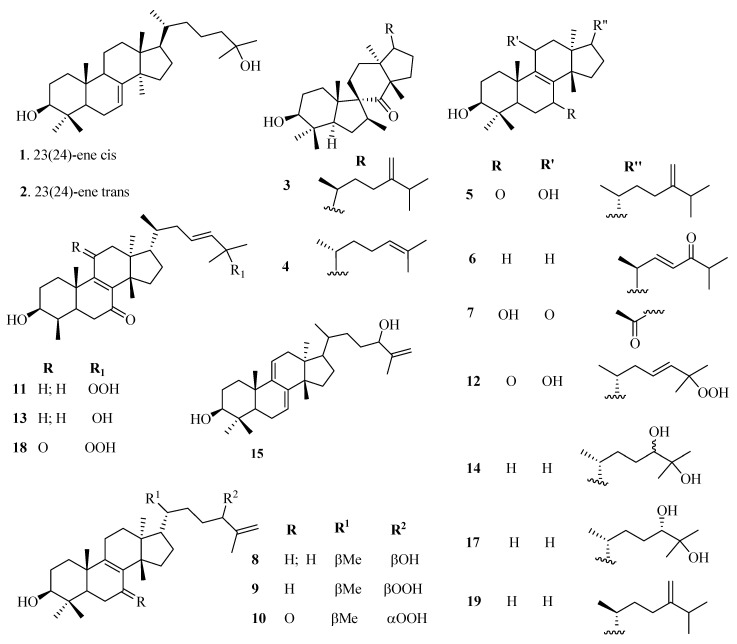
Chemical structures of tirucallane triterpenoids isolated from genus *Euphorbia.*

**Figure 2 molecules-25-04019-f002:**
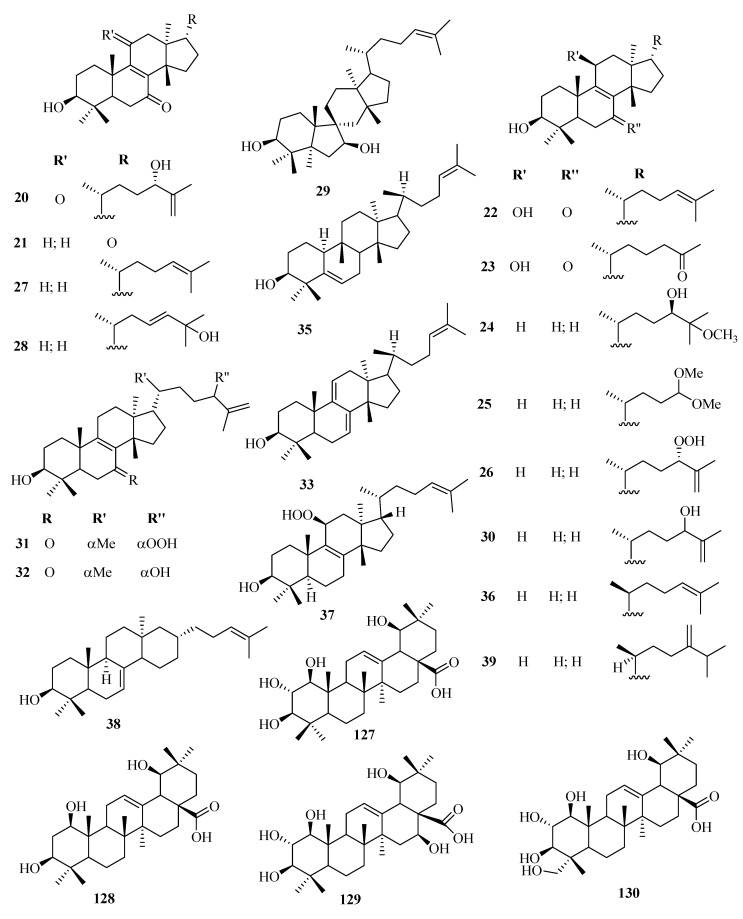
Structures of Euphane and some oleanane triterpenoids isolated from genus *Euphorbia.*

**Figure 3 molecules-25-04019-f003:**
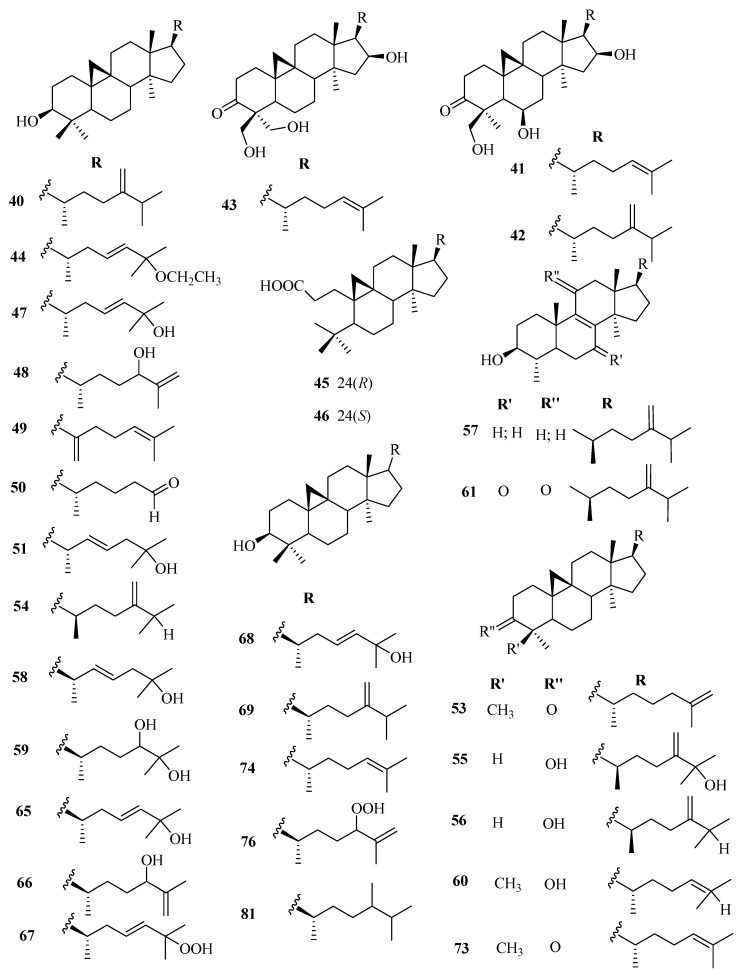
Chemical structures of cycloartane triterpenoids isolated from genus *Euphorbia.*

**Figure 4 molecules-25-04019-f004:**
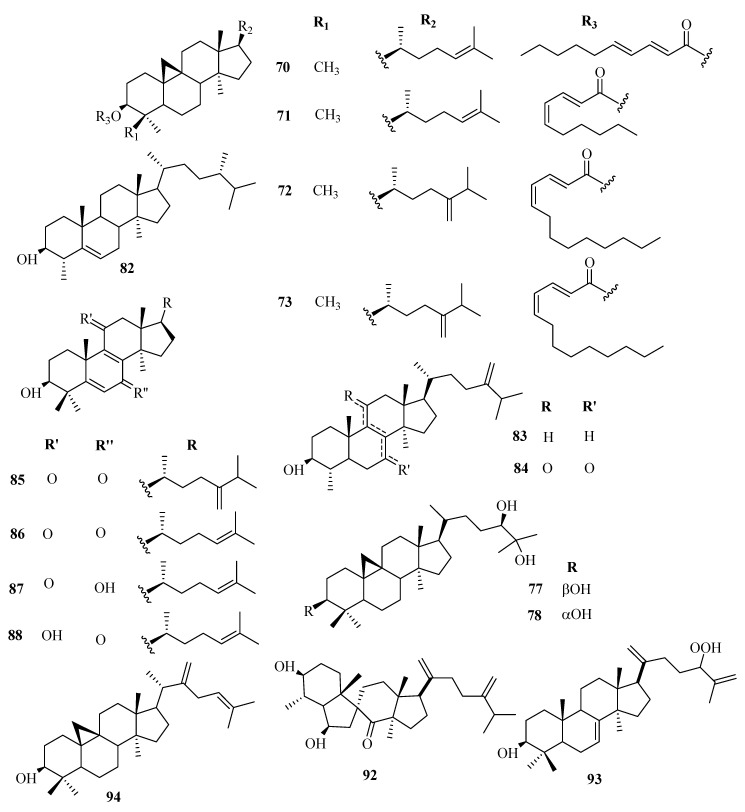
Structures of cycloartane and lanostane triterpenoids isolated from genus *Euphorbia.*

**Figure 5 molecules-25-04019-f005:**
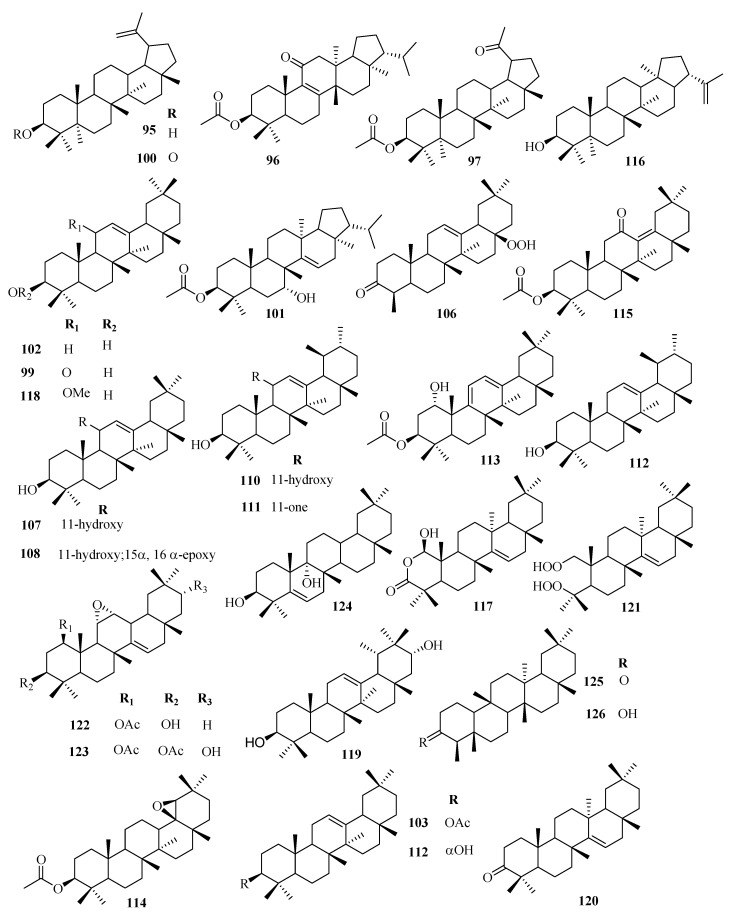
Chemical structures of pentacyclic triterpenoids isolated from genus *Euphorbia.*

**Table 1 molecules-25-04019-t001:** Traditional uses and biological activities reported for *Euphorbia* species.

Species	Part Used	Medicinal Uses	Biological Activities	References
*E. antiquorum*	Latex, stems, roots	Abnormalities, fuelwood, toothache, nervous system disorders, muscular-skeletal disorders	Antiviral, anti-inflammatory, cytotoxicity	[14,22]
*E. apios*	Roots, stems	Skin irritation	Laxative	[28]
*E. arbuscula*	Whole plant	Social events, ornamental	Wound healing	[14]
*E. argillicolq*	Latex	Vulnerary wounds	Anti-inflammation	[29]
*E. armena*	Latex	Healing wounds	Wound healing, constipation	[29]
*E. atoto*	Latex, stems	Pain reliever, muscular-skeletal system disorder	Antifungal	[14]
*E. caducifolia*	Stems, roots	Fuels, vertebrate poisoning	Unspecified	[14]
*E. candelabrum*	Whole plant	Materials/fuel woods	Unspecified	[14]
*E. chamaesyce*	Whole plant	Mental disorders	Antidepressant	[14]
*E. coniosperma*	Latex, aerial parts	Wounds, snake bites, enuresis, warts, scabies	Antifungal, antibacterial	[30]
*E. cooperi*	Stems, latex, roots	Fishing, skin irritant, nervous system syndrome	Antibacterial, antidepressant	[14,21]
*E. corollata*	Latex	Hemorrhages	Antiangiogenic	[14]
*E. cotinifolia*	Latex, stems	Vertebrate poisoning	Antiviral, ant inflammation	[14]
*E. denticulata*	Stems, roots, flowers	Obesity, diabetes,	Antibacterial, antioxidant	[31]
*E. dracunculoides*	Latex	Laxative, diuretic	Analgesic	[14,28]
*E. duseimata*	Stems, latex, roots	Mental illness	Antidepressant	[23]
*E. esula*	Stems, roots	Cancer, warts, inflammation, swelling	Antiviral, ant inflammation, anticancer	[14]
*E. falcata*	Roots, latex	Skin irritation & eczema	Antifungal activities	[29]
*E. fischeriana*	Roots	Neoplasms	Anti-bacterial,	[14,19]
*E. genistoides*	Whole plant	Metabolic system disorders	Diaphoretic properties	[14]
*E. helioscopia*	Latex, stems	Non vertebrate poisoning, food	Antihistaminic properties	[14,32,33]
*E. heteradena*	Stems, latex	Constipation	Constipation	[29]
*E. heterophylla*	Flowers, latex	Bees plant	Antifungal	[14]
*E. hirta*	Latex, stems, roots	Dysentery, bronchitis, pain, infestations (scabies, lies), poisoning, endocrine disorders	Anti-inflammation, antimalarial, anticancer,	[14,18,34]
*E. hispida*	Whole plant	Animal food	Anti-diarrhea	[14]
*E. humifusa*	Whole plant, latex	Cancer, influenza	Anticancer, anti-influenza	[35]
*E. inaequilatera*	Latex, stems	Circulatory system disorders, food	Antioxidant	[14]
*E. ingens*	Bark, latex	Cancerous lesion, eye tumor, snake bites, mental disorders	Anticancer, neoplasms	[14,21,23]
*E. lathyris*	Latex, roots, stems	Digestive disorders (purgative, emetic, diarrhea, constipation), blood purifier	Anticancer, antiviral, ant-inflammation	[14]
*E. lagascae*	Whole plant, latex	Rheumatism, bladder stones.	Anti-proliferative; antibacterial	[16]
*E. macroclada*	Latex	Warts, wound, snake-scorpion bites, tumors, toothache, eczema,	Antipyretic in malaria, antihaemorrhoidal, antifungal	[29,36,37]
*E. maculata*	Latex, stems, roots	Injuries, skin/subcutaneous cellular tissue disorders	Anti-inflammation, antifungal	[14,38]
*E. matabelensis*	Whole plant	Materials/fuel woods	Unspecified	[14]
*E. milii*	Whole plant, stem, roots	Ornamental, diarrhea in cattle, detoxicant, cancer, hepatitis, tumors	Anticancer, antiviral	[23,26,36]
*E. misera*	Stems, roots	Stomachache, constipation	Antidepressant	[14]
*E. myrsinites*	Aerial, latex	Stomach worms	Anti-intestinal, Anthelmintic	[28]
*E. neriifolia*	Latex, whole plant	Abdominal pains, bronchitis, tumors, leukoderma, piles, anemia, fever, respiratory	Anti-inflammatory, anticancer, Anti-bacterial	[39]
*E. nicaeensis*	Latex, aerial parts	Eczema, healing wounds	Anthelmintic, antifungal	[29]
*E. nivulia*	Whole plant, latex, stems	Enteritis, malaria, wounds, scabies, lies, mange, dysentery, poisonous snake bites, mental disorders	Anti-influenza, antifungal, antibacterial	[14,23,35]
*E. parviflora*	Latex, stems, roots	Respiratory disorders, asthma, coryza, anticatarrhal, snake bites	Antimalarial, anti-influenza, antibacterial, anti-inflammatory, anti-asthmatic	[14,16]
*E. pekinensis*	Whole plant, latex, stems	Abnormalities, non-vertebrate poisoning	Anti-inflammatory, cytotoxicity	[14]
*E. peplus*	Latex/sap	Cancer, purgative, warts, corns, asthma, bronchitis	Anticancer, anti-inflammation	[14,27]
*E. prostrata*	Latex, roots	Genitourinary disorders, endocrine system disorders, diabetes	Inflammatory, cytotoxicity	[14,23]
*E. pulcherrima*	Whole plant	Environmental	Anti-Alzheimer	[14,40]
*E. qarad*	Stems, roots	Diabetes, mental illness	Antidepressant	[23]
*E. resinifera*	Latex	Healing wounds	Antibacterial	[41]
*E. rigida*	Latex	Asthma	Anthelmintic, antihemorrhoidal	[28]
*E. royleana*	Whole plant	Asthma, jaundice, cough	Anti-HIV, antibacterial	[42]
*E. seguieriana*	Latex, aerial	Inflamed wounds, malaria, constipation	Antipyretic in malaria, anti-inflammatory	[29]
*E. serrata*	Latex	Curdle milk, food additive	Antihistaminic	[14,43]
*E. sessiliflora*	Stems, roots	Diarrheal, edema, ascites, cancer, gonorrhea	Antiviral, anticancer	[44]
*E. sikkimensis*	Roots, stems	Swellings, tumor, malaria, bacillary, jaundice, dysentery, rheumatism	Antimalarial, anticancer	[45,46]
*E. stracheyi*	Roots, latex	Hemostasis, analgesia	Antiangiogenic	[47]
*E. stricta*	Latex stems, roots	Cleaning water, toothache, laxative, diuretic	Analgesic in toothache, callus	[14,28,48]
*E. thymifolia*	Leaves, seeds, roots	Treating worms in children, stimulant, laxative, hematuria, virginal bleeding, wounds, carbuncles, epistaxis	Antiviral, antimalarial, anticancer	[19,38]
*E. tirucalli*	Whole plant, latex	Vertebrate poison, fuel wood, snake bite, jaundice,	Anticancer, antimodulatory	[14,16,23,49,50,51]
*E. virgata*	Stems, flowers	Eczema	Antifungal	[29]
*E*. *Wallichii*	Latex, Stem, roots	Constipation	Constipation	[52]

**Table 2 molecules-25-04019-t002:** Triterpenoids isolated from the genus *Euphorbia.*

Type	No	Compound Name	Occurrence (Solvent, Part Used)	Biological Effect (Cell Type)	References
Tirucallane	1	(23*E*)-3β-25-dihydroxytirucalla-7,23-diene	*E. maculata* (methanol, whole plant)	Anti-inflammatory	[58]
	2	(23*Z*)-3β-25-dihydroxy-tirucalla-7,23-diene	*E. maculata* (methanol, whole plant)	Anti-inflammatory	[58]
	3	euphorol K	*E. resinifera* (methanol, roots)	Cytotoxic (MCF-7, C6)	[59]
	4	kansuinone	*E. resinifera* (methanol, roots)	Cytotoxic (MCF-7, C6)	[59]
	5	euphorol E	*E. resinifera* (methanol, roots)	Cytotoxic (MCF-7, C6)	[60]
	6	euphorol F	*E. resinifera* (methanol, roots)	Cytotoxic (MCF-7, C6)	[60]
	7	euphorol G	*E. resinifera* (methanol, roots)	Cytotoxic (MCF-7, C6)	[60]
	8	(–)-(24*R*)-tirucalla-8,25-diene-3*β*,24-diol	*E. micractina* (ethanol, roots)	Cytotoxic (A2780)	[61]
	9	(–)-(24*R*),24-hydroperoxytirucalla-8,25-dien-3*β*-ol-7-one	*E. micractina* (ethanol, roots)	Cytotoxic (A2780)	[61]
	10	(–)-(24*S*),24-hydroperoxytirucalla-8,25-dien-3*β*-ol-7-one	*E. micractina* (ethanol, roots)	Cytotoxic (A2780)	[61]
	11	(–)-(23*E*),25-hydroperoxytirucalla-8,23-dien-3*β*-ol-7-one	*E. micractina* (ethanol, roots)	Cytotoxic (A2780)	[61]
	12	euphorol L	*E. tirucalli* (whole plant, ethanol)	Cytotoxic (K562, MCF-7)	[62]
	13	(–)-(23*E*)-tirucalla-8,23-diene-3*β*,25-diol-7-one	*E. micractina* (ethanol, roots)	Cytotoxic (A2780)	[61]
	14	euphorol M	*E. tirucalli* (whole plant, ethanol)	Cytotoxic (K562, MCF-7)	[62]
	15	(–)-(24*R*)-tirucalla-7,9(11),25-triene-3*β*,24-diol	*E. micractina* (ethanol, roots)	Cytotoxic (A2780)	[61]
	16	tirucall-spiro [5,6]-24-methylene-3*β*,7*β*-diol-8-one	*E. resinifera* (methanol, roots)	Cytotoxic (MCF-7, C6)	[59]
	17	euphorol N	*E. tirucalli* (whole plant, ethanol)	Cytotoxic (K562, MCF-7)	[62]
	18	(+)-(23*E*),25-hydroperoxytirucalla-8,23-dien-3*β* -ol-7,11-dione	*E. micractina* (ethanol, roots)	Cytotoxic (A2780)	[61]
	19	24-methyltirucalla-8,24(24′)-dien-3*β*-ol	*E. antiquorum* (Latex)	Inhibitory (EBV-EA)	[63]
Euphane	20	(24*R*)-eupha-8,25-diene-3β,24-diol-7,11-dione	*E. resinifera* (methanol, roots)	Cytotoxic (MCF-7, C6)	[60]
	21	4,4,14β,-trimethyl-5α-pregn-8-ene-3β,17α-diol-7-one	*E. maculata (n*-hexane)	Anti-inflammatory	[58,60]
	22	eupha-8,24-diene-3β,11β-diol-7-one	*E. resinifera* (methanol, roots)	Anti-inflammatory, cytotoxic (MCF-7, C6)	[58,60]
	23	eupha-8-ene-3β,11β-diol-7,24-dione	*E. resinifera* (methanol, roots)	Cytotoxic (MCF-7, C6)	[60]
	24	(24 *R*)-eupha-25-methoxyl-8-ene-3β,24-diol	*E. resinifera* (methanol, roots)	Cytotoxic (MCF-7, C6)	[60]
	25	24,24-dimethoxy-25,26,27-trinoreuphan-3β-ol	*E. antiquoru* (acetone, stems)	Anti-HIV(HIV-1)	[64]
	26	(24*S*)-24-hydroperoxyeupha-8,25-dien-3β-ol	*E. antiquoru* (acetone, stems)	Anti-HIV(HIV-1)	[64]
	27	kansenone, (20*R*,23*E*)-eupha-8,23-diene-3β,25-diol	*E. resinifera* (methanol, roots)	Cytotoxic (MCF-7, C6)	[59,60]
	28	kansenonol,11-oxo-kansenonol, kansenol	*E. resinifera* (methanol, roots)	Cytotoxic (MCF-7, C6)	[60]
	29	kansuinone	*E. retusa*, *E. resinifera* (methanol, roots)	Cytotoxic (11β-HSD1), anti-inflammatory	[56,58]
	30	(24*R*)-eupha-8,25-diene-3β,24-diol	*E. resinifera* (methanol, roots)	Cytotoxic (MCF-7, C6)	[60]
	31	(+)-(24*S*),24-hydroperoxyeupha-8,25-dien-3*β*-ol-7-one	*E. micractina* (ethanol, roots)	Cytotoxic (A2780)	[61]
	32	(+)-(24*S*)-eupha-8,25-diene-3*β*,24-diol-7-one	*E. micractina* (ethanol, roots)	Cytotoxic (A2780)	[61]
	33	eupha-7,9(11),24-trien-3*β*-ol (antiquol C)	*E. antiquorum* (latex)	Inhibitory (EBV-EA)	[63]
	34	23(*E*)-eupha-8,23–diene-3β,25-diol-7-one	*E. neriifolia* (methanol, roots)	Antiangiogenic	[65]
	35	19(10, 9) *abeo*-8α,9*β*,10α-eupha-5,24-dien-3*β*-ol (antiquol B)	*E. antiquorum* (latex)	Inhibitory (EBV-EA)	[63]
	36	Euphol	*E. antiquorum* (latex)	Inhibitory (EBV-EA)	[63]
	37	11*β*-hydroperoxyeupha-8,24-diene-3*β*-ol	*resinifera* (methanol, roots)	Cytotoxic (MCF-7, C6)	[59]
	38	lemmaphylla-7, 21-dien-3*β*-ol	*E. antiquorum* (latex)	Inhibitory (EBV-EA)	[63]
	39	24-methyleupha-8,24(24^1^)-dien-3*β*-ol	*E. antiquorum* (latex)	Inhibitory (EBV-EA)	[63]
Cycloartane	40	24-methylenecycloartane	*E. neriifolia* (ethyl acetate, leaves), *E. schimperi* (methanol, aerial), *E. pterococca* (acetone, whole plant)	Cytotoxic (MCF-7, C6), antiangiogenic, cytotoxic (MCF-7, HepG2, HCT-116), inhibitory (α/β-hydrolase 12)	[60,61,62,63,64,65,66,67,68]
	41	neriifolins A	*E. neriifolia* (ethyl acetate, leaves)	Cytotoxic (MCF-7)	[66]
	42	neriifolins B	*E. neriifolia* (ethyl acetate, leaves)	Cytotoxic (MCF-7)	[66]
	43	neriifolins C	*E. neriifolia* (ethyl acetate, leaves)	Cytotoxic (MCF-7)	[66]
	44	23(*E*)-cycloart-23-en-25-ethoxy-3β-ol	*E. humifusa* (ethanol, whole plant)	Cytotoxic (SGC-7901)	[69]
	45	24(*R*)-3,4-*seco*cycloart-4(29),25-dien-24-hydroxy-3-oic acid	*E. humifusa* (ethanol, whole plant)	Cytotoxic (SGC-7901)	[69]
	46	24(*S*)-3,4-*seco*-cycloart-4(29),25-dien-24-hydroxy-3-oic acid	*E. humifusa* (ethanol, whole plant)	Cytotoxic (SGC-7901)	[69]
	47	23(*Z*)-cycloart-23-en-3β,25-diol	*E. humifusa* (ethanol, whole plant)	Cytotoxic (SGC-7901)	[69]
	48	24(*S*)-cycloart-25-en-3β,24-diol	*E. humifusa* (ethanol, whole plant)	Cytotoxic (SGC-7901)	[69]
	49	cycloschimperols A (cycloart-20,24-dien-3β-ol)	*E. schimperi* (methanol, aerial)	Cytotoxic (MCF-7, HepG2, HCT-116)	[67]
	50	cycloschimperols B (26,27-dinor-3β-hydroxy cycloartan-25-al)	*E. schimperi* (methanol, aerial)	Cytotoxic (MCF-7, HepG2, HCT-116)	[67]
	51	cycloart-(23*Z*)-ene-3α,25-diol	*E. pulcherrima* (acetone, aerial)	Inhibitory (osteoclastogenesis)	[55]
	52	cycloart-5-ene-3β,25-diol	*E. neriifolia* (methanol, roots)	Antiangiogenic	[65]
	53	cycloart-25-en-3-one	*E. schimperi* (methanol, aerial)	Cytotoxic (MCF-7, HepG2, HCT-116)	[67]
	54	24-methylene cycloartanol	*E. clementei* (chloroform, roots)	Inhibitory (K562, HL60)	[70]
	55	24-methylene cycloartan-3*β*,25-diol	*E. clementei* (chloroform, roots)	Inhibitory (K562, HL60)	[70]
	56	cycloeucalenol	*E. clementei* (chloroform, roots), *E. maculata* (methanol, whole plant), *E. pterococca* (acetone, whole plant), *E. pulcherrima* (acetone, aerial)	Inhibitory (osteoclastogenesis), anti-inflammatory, inhibitory (α/β-hydrolase 12), inhibitory (K562, HL60)	[55,58,68,70]
	57	obtusifoliol	*E. clementei* (chloroform, roots); *E. maculata* (methanol, whole plant), *E. pterococca* (acetone, whole plant)	Anti-inflammatory, inhibitory (α/β-hydrolase 12), inhibitory (K562, HL60)	[58,68,70]
	58	cycloart-22*E*-ene-3*β*,25-diol	*E. clementei* (chloroform, roots)	Inhibitory (K562, HL60)	[70]
	59	(3*β*,9*β*,24*R*) 9,19-cyclolanostane-3,24,25-triol-1-methyl-cyclobutene	*E. clementei* (chloroform, roots)	Inhibitory (K562, HL60)	[70]
	60	cycloartenyl acetate	*E. clementei* (chloroform, roots), *E. pterococca* (acetone, whole plant)	Inhibitory (α/β-hydrolase 12), inhibitory (K562, HL60)	[68,70]
	61	7,11-dioxoobtusifoliol	*E. pedroi* (methanol, roots)	Cytotoxic (Colo320)	[53]
	62	cycloart-23(*E*)-ene-3β,25-diol	*E. pedroi* (methanol, roots), *E. macrostegia* (chloroform, whole plant)	Cytotoxic (Colo320), cytotoxic (MCF-7)	[53,71]
	63	cycloart-25-ene-3β,24-diol	*E. pedroi* (methanol, roots), *E. macrostegia* (chloroform, whole plant)	Cytotoxic (Colo320), cytotoxic (MCF-7)	[53,71]
	64	3-hydroxycycloart-25-ene-24-hydroperoxide	*E. maculata* (methanol, whole plant)	Anti-inflammatory	[58]
	65	3β-hydroxy-26-nor-9,19-cyclolanost-23-en-25-one	*E. maculata* (methanol, whole plant)	Anti-inflammatory	[58]
	66	cycloart-23-en-3β,25-diol,	*E. maculata* (methanol, whole plant), *E. macrostegia* (chloroform, whole plant)	Anti-inflammatory, cytotoxic (MCF-7)	[58,71]
	67	25-hydroperoxycycloart-23-en-3β-ol	*E. hirta* (dichloromethane, stems)	Cytotoxic (A549, HCT)	[72]
	68	cycloart-23(*Z*)-ene-3β,25-diol	*E. macrostegia* (chloroform, whole plant)	Cytotoxic (MCF-7)	[71]
	69	cycloart-24-en-3β-ol	*E. macrostegia* (chloroform, whole plant)	Cytotoxic (MCF-7)	[71]
	70	cycloartenyl-2′*E*,4′*E*-decadienoate	*E. pterococca* (acetone, whole plant)	Inhibitory (α/β-hydrolase 12)	[68]
	71	cycloartenyl-2′*E*,4′*Z*-decadienoate	*E. pterococca* (acetone, whole plant)	Inhibitory (α/β-hydrolase 12)	[68]
	72	24-methylenecycloartanyl-2′E,4′*Z*-tetradecadienoate	*E. pterococca* (acetone, whole plant)	Inhibitory (α/β-hydrolase 12)	[68]
	73	24-oxo-29-norcycloartanyl-2′*E*,4′*Z*-hexadecadienoate	*E. pterococca* (acetone, whole plant)	Inhibitory (α/β-hydrolase 12)	[68]
	74	cycloartenol	*E. pterococca* (acetone, whole plant), *E. hirta* (dichloromethane, stems)	Inhibitory (α/β-hydrolase 12)	[68,72]
	75	cycloartenone	*E. pterococca* (acetone, whole plant)	Inhibitory (α/β-hydrolase 12)	[68]
	76	24-hydroperoxycycloart-25-en-3β-ol	*E. hirta* (dichloromethane, stems)	Cytotoxic (A549, HCT)	[72]
	77	(24*R*)-cycloartane-3β,24,25-triol	*E. pulcherrima* (acetone, aerial)	Inhibitory (osteoclastogenesis)	[55]
	78	(24*R*)-cycloartane-3α,24,25-triol	*E. pulcherrima* (acetone, aerial)	Inhibitory (osteoclastogenesis)	[55]
	79	3*β*-hydroxy-24-methylene-9,19-cyclolanostane	*E. retusa* (methanol, whole plant)	Inhibitory (osteoclastogenesis)	[73]
	80	24-methylenecycloartanol	*E. retusa* (methanol, whole plant)	Cytotoxic (HT-3)	[73]
	81	3*β*-hydroxy-9, 19-cyclolanostane (cyclolaudanol)	*E. retusa* (methanol, whole plant)	Cytotoxic (HT-3)	[73]
	82	3*β*,24*S-*ergost-5-en-ol	*E. retusa* (methanol, whole plant)	Cytotoxic (HT-3)	[73]
	83	3β-hydroxy-4α,14α-dimethyl-5α-ergosta-7,9(11), 24(28)-trien.	*E. humifusa* (ethanol, whole plant)	Cytotoxic (SGC-7901)	[69]
	84	3β-hydroxy-4α,14α-dimethyl-5α-ergosta-8,24(28)-dien-7,11-one	*E. humifusa* (ethanol, whole plant)	Cytotoxic (SGC-7901)	[69]
Lanostane	85	(3β)-3-hydroxy-24-methylenelanost-8-ene-7,11-dione	*E. humifusa* (methanol, whole plant)	Not evaluated	[74]
	86	(3β)-3-hydroxylanosta-8,24-diene-7,11-dione	*E. humifusa* (methanol, whole plant)	Not evaluated	[74]
	87	(3β,7α)-3,7-dihydroxylanosta-8,24-dien-11-one,	*E. humifusa* (methanol, whole plant)	Not evaluated	[74]
	88	(3β,11β)-3,11-dihydroxylanosta-8,24-dien-7-one	*E. humifusa* (methanol, whole plant)	Not evaluated	[74]
	89	3β, 24, 25-trihydroxylanost-8-en-11-one	*E. humifusa* (methanol, whole plant)	Not evaluated	[74]
	90	3*β*-hydroxy-25-methyloxylanosta-8, 23-diene	*E. pekinensis* (methanol, whole plant)	Not evaluated	[75]
	91	3*β*, 25-dihydroxylanosta-8,23-diene	*E. pekinensis* (methanol, whole plant)	Not evaluated	[75]
	92	(3*S*,4*S*,7*S*,9R)-4-methyl-3,7-dihydroxy-7(8→9) abeo-lanost-24(28)-en-8-one	*E. maculata* (methanol, whole plant)	Anti-inflammatory	[58]
	93	24-hydroperoxylanost-7,25-dien-3β-ol,	*E. maculata* (methanol, whole plant)	Anti-inflammatory	[58]
	94	9,19-cyclolanost-22(22′),24-diene-3β-ol (nerifoliene)	*E. nerifolia* (fresh latex)	Not evaluated	[76]
Lupane	95	lupeol	*E. hirta* (dichloromethane, stems), *E. latifolia, E. maculata*, (methanol, whole plant)	Anti-inflammatory, cytotoxic (A549, HCT)	[58,72,77]
	96	euphorimaoid A.	*E. pulcherrima* (methanol, whole plant)	Inhibitory (osteoclastogenesis)	[55]
	97	3β-acetoxy-30-nor-20-oxolupane	*E. pulcherrima* (methanol, whole plant)	Inhibitory (osteoclastogenesis)	[55]
	98	lupenyl acetate, lupeol palmitate,	*E. segetalis*, *E. chamaesyce*, *E. helioscopia*, (whole plant)	Inhibitory (osteoclastogenesis), cytotoxic (HeLa)	[55,78]
	99	3β-acetoxyolean-12-en-11-one	*E. pulcherrima* (methanol, whole plant)	Inhibitory (osteoclastogenesis)	[55]
	100	lupenone	*E. segetalis*, *E. chamaesyce*, *E. helioscopia*, (methanol, whole plant)	Inhibitory (osteoclastogenesis), cytotoxic (HeLa)	[55,78]
	101	euphorimaoid A.	*E. pulcherrima* (methanol, whole plant)	Inhibitory (osteoclastogenesis)	[55]
Oleanane	102	β-amyrin	*E. hirta* (ethanol, roots)	Cytotoxic (A549, HCT)	[72]
	103	β*-*amyrin acetate	*E. pedroi*, *E. nematocypha* (methanol, roots)	Cytotoxic (Colo320)	[53,77]
	104	β-amyrin palmitate	*E. pedroi*, *E. nematocypha* (methanol, roots)	Cytotoxic (Colo320)	[53,77]
	105	β-amyrin benzoate	*E. pedroi* (methanol, roots)	Cytotoxic (Colo320)	[53,77]
	106	oleanolic acid	*E. pedroi*, *E. nematocypha*, *E. alatavica* (methanol, roots)	Cytotoxic (Colo320)	[53,77]
	107	12-oleanene-3β-11-diol	*E. maculata* (methanol, roots)	Anti-inflammatory	[58]
	108	(3β,15α,16α)-15,16-epoxy-olean-12-en-3β-ol	*E. maculata* (methanol, roots)	Anti-inflammatory	[58]
	109	3*β*,23-dihydroxy-12-oleanan-28-oic acid	*E. alatavica* (methanol, roots)	Cytotoxic (HeLa, MCF-7, A549)	[57]
	110	urs-12-ene-3β,11α-diol	*E. maculata* (methanol, roots)	Anti-inflammatory	[58]
	111	neoilexonol	*E. maculata* (methanol, roots)	Anti-inflammatory	[58]
	112	α-amyrin	*E. hirta* (ethanol, roots)	Cytotoxic (A549, HCT116)	[72]
	113	1α-hydroxy-3β-acetoxy-olean-9,12-diene,3β-acetyloxy-olean-13(18)-en-12-one	*E. pulcherrima* (acetone, aerial plant)	Inhibitory (osteoclastogenesis)	[55]
	114	18,19-epoxyolean-3β-ol acetate	*E. pulcherrima* (acetone, aerial plant)	Inhibitory (osteoclastogenesis)	[55]
	115	3β-acetoxyolean-12-en-11-one	*E. pulcherrima* (acetone, aerial plant)	Inhibitory (osteoclastogenesis)	[55]
	116	3β-hydroxyhop-22(29)-ene	*E. pulcherrima* (acetone, aerial plant)	Inhibitory (osteoclastogenesis)	[55]
	117	3,2-seco-taraxer-14-en-3,2-lactone	*E. alatavica* (methanol, roots)	Cytotoxic (HeLa, MCF-7, A549)	[57]
	118	olean-12-ene-11α-methoxy-3β-acetate	*E. pulcherrima* (acetone, aerial plant)	Inhibitory (osteoclastogenesis)	[55]
Taraxarane	119	taraxast-12-ene-3β,20,21(α)-triol	*E. denticulate* (acetone, aerial plant)	Cytotoxic (DU-145)	[79]
	120	taraxerone	*E. alatavica* (methanol, roots), *E. hirta* (ethanol, roots)	Cytotoxic (A549, HCT)	[57,72]
	121	2,3-*seco*-taraxer-14-en-2,3-bioic acid	*E. alatavica* (methanol, roots)	Cytotoxic (HeLa, MCF-7, A549)	[57]
	122	1β-acetoxy-3β-hydroxy-11α,12α-oxidotaraxer-14-ene	*E. geniculata* (ethyl acetate, aerial plant)	Not evaluated	[80]
	123	1β,3β-diacetoxy-21α-hydroxy-11α,12α-oxidotaraxer-14-ene	*E. geniculata* (ethyl acetate, aerial plant)	Not evaluated	[80]
	124	3β,9α,20α-trihydroxy-Ψ-taraxast-5-ene	*E. geniculata* (ethyl acetate, aerial plant)	Not evaluated	[80]
Friedelane	125	friedelin	*E. geniculata* (ethyl acetate, aerial plant)	Not evaluated	[80]
	126	friedelinol	*E. geniculata* (ethyl acetate, aerial plant)	Not evaluated	[80]
Other oleanane	127	(1β,2α,3β,19β)-1,2,3,19-tetrahydroxyolean-12-en-28-oic acid	*E. sieboldiana* (ethanol, whole plant)	Cytotoxic (HeLa, Hep-G2)	[7]
	128	(1β,3β,19β)-1,3,19-trihydroxyolean-12-en-28-oic acid	*E. sieboldiana* (ethanol, whole plant)	Cytotoxic (HeLa, Hep-G2)	[7]
	129	(1β,2α,3β,16β,19β)-1,2,3,16,19-pentahydroxyolean-12-en-28-oic acid	*E. sieboldiana* (ethanol, whole plant)	Cytotoxic (HeLa, Hep-G2)	[7]
	130	(1β,2α,3β,19β,23)-1,2,3,19,23-pentahydroxyolean-12-en-28-oic acid	*E. sieboldiana* (ethanol, whole plant)	Cytotoxic (HeLa, Hep-G2)	[7]

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
