# Peer review of "A Review of the Ethnomedicinal Uses, Biological Activities, and Triterpenoids of *Euphorbia* Species"

_molecules, 2020, doi:10.3390/molecules25174019_

Round 1
Reviewer 1 Report
The review is very interesting and well done. I suggesto to arrange the plant list (table 1) in alphabetical order. It is not clear what order has been used by the authors and certainly it is not easy to read names in a random order.
My only concern is that the autors do not quote a recent review (Biomolecules 2019, 9, 337; doi:10.3390/biom9080337). I understand that the present paper is focused on tripterpenoids, but the title refers to the genus as a whole. I strongly suggest to quote and discuss this missing reference (I am not involved in any way in that paper!!!!).
Reviewer 2 Report
Dear Author,
I have revised the manuscript entitled "A review of the ethnomedicinal uses, phytochemistry and biological activities of Euphorbia species". The manuscript is a review of the state of the art of triterpenes that have been isolated from species of the genus Euphorbia. Although the title of the manuscript highlights the ethnomedicinal uses, phytochemistry and biological activities, in the review the authors describe some ethnobotanical uses, well included in Table 1, but they limit their review to describe only the chemistry of triterpenes as well as their effects as "anticancer" compounds. The title can confuse the reader, so I suggest that the authors modify their title to avoid confusion because only chemistry and anticancer activities of triterpenoids are included.
On the other hand, it is recommended to deepen the analysis of triterpenes and their activities as anticancer agents. This review is limited to describe some biological effects of some triterpenoids, but many considerations could be done regarding the chemistry of triterpenoids, like the skeleton type, oxidation pattern, substituyent pattern among other, with their biological activities. It is also recommended to modify Table 2 to place the biological activity described for each compound if this information exists, since it is very difficult to track biological activity data in the text.
Reviewer 3 Report
The manuscript “A review of the ethnomedicinal uses, phytochemistry and biological activities of Euphorbia species” was submitted to Molecules for publication. The article gives an overview on the various triterpenoids isolated from genus Euphorbia along with their effects on cancer cell lines and other pharmacological targets.
Broad comments:
The topic of the review and the therein presented species are interesting and worth to be reviewed. Especially the high number of structurally diverse triterpenoids isolated from the genus Euphorbia is fascinating. The article covers a high number of species and compounds along with some interesting activities and does this in a good and readable way. However, regarding the appearance of the article some corrections must be made as it appears in some parts a little bit sloppy.
- Literature survey databases: Here, the authors must more precisely describe how the literature was retrieved, e.g. which keywords were used in which database and how the search was carried out. Same as for a research article the reader should be able to obtain the same results as the authors. Only giving the databases used therefor is not sufficient.
- Ethnomedical uses of Euphorbia species: In this chapter and in table 1 several Euphorbia species are listed along with the parts used, their medicinal uses and biological activities. This is all fine, but some of the species names are not spelled correctly and/or were meanwhile reclassified. For example, E. legascae is spelled E. lagascea and E. pilulifera is meanwhile named E. parviflora. E. Emadi contains several mistakes. First, the species name must be written in small letters, second, it is spelled E. emodi, and third, is was reclassified as E. hispida. It is also not clear why E. stricta and E. dracunculoide (which is spelled dracunculoides) are listed together even though they are two different species and why E. falcata subsp. falcata var. falcata was presented down to a variety level while many other accepted subspecies and varieties were not mentioned (e.g. subspecies of E. dracunculoides). As this review article only covers one genus (though a rather big one), more attention should be paid to the taxonomic correctness and actuality. Thus, all species should be checked with theplantlist and corrected if need be. The former names can be indicated in parenthesis or if they are numerous also presented in an additional column. Another point is the conciseness of the taxonomy. It must be chosen if the author names should be indicated in the species names or not. This is up to the authors of the review but should be uniform throughout the manuscript. After the corrections were done, the species should be listed in alphabetic order, so it is easier for the reader to find a species and the respective reference.
- Isolated triterpenoids: Because this review only covers one genus, Euphorbia should be abbreviated with E. after its first appearance (which already happened in the introduction). Regarding the compound names, also in this chapter uniformity is missing. Compound names are alternatively written in big or small starting letters in table 2 and in the manuscript. In English language compound names are usually written in small starting letters and should be spelled so in this manuscript. However, if the authors prefer not to do so, they should then see that all compound names are written in the same style. This accounts for both, systematic and trivial names of the compounds, and for the names of solvents and plant parts, respectivey.
Specific comments:
Line 18-20: This sentence is not understandable and must be corrected/rewritten.
Line 237: It should read “Fangfang et al.”
Line 253: It should read “Cytotoxic effects”
Line 279: A space is missing in this line.
Line 337: It should read “with an IC50 of 8.07 µg/mL”
Please align the abbreviations.
Round 2
Reviewer 2 Report
The authors took into account the recommendations made previously. I recommend the publication of the manuscript.